# A Randomized Cross-Over Study Comparing Cooling Methods for Exercise-Induced Hyperthermia in Working Dogs in Training

**DOI:** 10.3390/ani13233673

**Published:** 2023-11-28

**Authors:** Sara C. Parnes, Amritha Mallikarjun, Meghan T. Ramos, Tesa A. Stone, Cynthia M. Otto

**Affiliations:** 1Penn Vet Working Dog Center, University of Pennsylvania School of Veterinary Medicine, Philadelphia, PA 19146, USA; scparnes@upenn.edu (S.C.P.); amritham@vet.upenn.edu (A.M.); tastone9@vet.upenn.edu (T.A.S.); cmotto@vet.upenn.edu (C.M.O.); 2Department of Clinical Sciences and Advanced Medicine, University of Pennsylvania, Philadelphia, PA 19104, USA

**Keywords:** hyperthermia, working canine, cooling, isopropyl alcohol, water immersion, heat stress, exercise

## Abstract

**Simple Summary:**

Working dogs are at high risk of heat stress and heat stroke, as they often are required to perform physically demanding tasks for long periods of time. Handlers’ intervention upon noticing signs of heat stress can prevent more serious consequences for the dog. This study assessed the effectiveness of two different, easily implementable cooling interventions that handlers could use with their dogs: partial water immersion and isopropyl alcohol application to paw pads. While both interventions were more effective than no intervention at all, partial water immersion cooled dogs faster and led to an overall cooler temperature in dogs post-exercise than isopropyl alcohol application. Isopropyl alcohol additionally raised dogs’ heart rates significantly more than partial water immersion or no intervention, which may be due to its aversive odor and potential for topical irritation. As such, partial water immersion is preferred over isopropyl alcohol for cooling dogs after exercise. Further studies are required to explore the extent to which dogs must be immersed in water and the effectiveness of other water-assisted cooling methods like cooling towels.

**Abstract:**

Working dogs are at a high risk of canine non-pyrogenic hyperthermia, a life-threatening condition that can occur due to physical exertion or environmental factors that inhibit dogs’ ability to cool themselves. Two frequently recommended cooling methods to reduce body temperature are water immersion and the application of isopropyl alcohol to paw pads. This cross-over study compared the relative efficacy of these methods in 12 working-dogs-in-training with post-exertional heat stress. On each study day, dogs had a physical exam and performed a warm-up exercise followed by sequential recalls in which dogs ran approximately 25 m between two designated handlers for 10 min until they showed multiple signs of heat stress or their core temperature reached 105 °F (40.6 °C). Dogs’ temperature and heart rate were collected after each recall. Dogs completed three study days, and each day, randomly received one of three interventions: passive cooling (no intervention), partial water immersion, or isopropyl alcohol. Post-intervention dogs rested for 20 min. Partial water immersion and isopropyl alcohol both cooled dogs more than no intervention, and water immersion cooled dogs more efficiently than isopropyl alcohol. Additionally, the application of isopropyl alcohol raised dogs’ heart rates more than water immersion or no intervention, suggesting that the process of applying isopropyl alcohol is potentially stressful to dogs. Thus, partial water immersion is preferred to cool dogs post-exertion due to its more efficient cooling and better tolerance of use.

## 1. Introduction

Canine non-pyrogenic hyperthermia is one of the greatest risks to working dogs’ daily health [1]. This condition occurs when a dog’s core body temperature increases beyond 103 °F (39.4 °C) due to physical exertion and/or environmental conditions that inhibit the dogs’ methods of cooling. Heat-induced injuries are the most common non-traumatic cause of death in military working dogs deployed to Iraq and Afghanistan [2] and the second most common cause of non-hostile action death in police canines [3]. Working dogs may be more susceptible to exertional non-pyrogenic hyperthermia as they operate at high levels of intensity for prolonged periods of time in a variety of climates [4]. The spectrum of canine hyperthermia progresses from heat stress (which encompasses initial physiologic responses to increased core temperature) to heat exhaustion (in which changes in physiologic function result in mild–moderate organ damage) to heat stroke (in which neurologic signs and severe organ damage occur and may result in death) [5]. Recent literature has demonstrated that core temperature alone does not dictate the stage of hyperthermia; working dogs often reach temperatures over 105 °F (40.5 °C) during normal exercise and activity [6,7,8,9]. As such, it is crucial to explore methods to accelerate cooling and reduce the risk of hyperthermia in working dogs.

Whether an individual dog exhibits heat stress or progresses to heat stroke is multifactorial and thought to be influenced by workload, ambient conditions, acclimatization, hydration, and owner/handler response [9,10,11]. Workload, ambient conditions, and acclimatization are not elements that can be well controlled in working dogs, as they must travel and perform physically challenging tasks in adverse environmental conditions (e.g., high temperatures). As such, preventing the progression of heat stress relies heavily on the owner/handler’s response to the dogs’ behavior and to preventatively hydrate and provide water to their dogs. Education of owners and handlers on methods to mitigate the risks and treat heat injury with effective cooling methods may be the most critical element in preventing canine hyperthermia injury and death. In one heat stroke study, dogs admitted to the Hebrew University Veterinary Teaching Hospital were more likely to survive when owners cooled their dogs within four hours prior to hospital admission. Dogs that were cooled at home had a lower rectal temperature on hospital admission, further supporting that cooling is beneficial to decreasing core body temperature [12]. As such, easy-to-implement cooling interventions that canine handlers can implement themselves are vital to maintaining their working dogs’ health.

Commonly discussed cooling methods for working dogs include both conductive methods (involving direct application of a cooling source such as immersion or application of water [13] or cooling vests [14]) and evaporative methods (transferring thermal energy by misting the patient with water or isopropyl alcohol and using a fan or natural air circulation to promote evaporation [15]), including the use of a cooling pad [16], a cooling vest, and/or the application of isopropyl alcohol to the foot pads. In this study, we examined one conductive method (partial water immersion) and one evaporative method (isopropyl alcohol application) as compared to passive cooling. Both of these methods are currently used in the field for working dogs experiencing heat stress [15]. There is a great deal of evidence from canine, human, and other non-human animal research suggesting that water immersion is an effective cooling technique [16,17,18,19,20]. Water immersion creates a thermal gradient between the dogs’ skin and the cooler water, transferring heat from the dog to the water, thus cooling the dog. In a recent canine exercise study comparing passive cooling and a chilled (39.2 °F, 4 °C) cooling mat, water immersion provided the most rapid cooling, even with ambient temperature water [16]. Full body immersion, however, may not be an option in deployment or field scenarios due to water availability and behavioral compliance from the dogs. A meta-analysis of human cooling methods post-hyperthermia suggests that water cooling rates are generally faster when the highest proportion of the body is immersed in water [21]; however, there are no direct comparisons of partial immersion to full body immersion, and no examination of partial water immersion in dogs.

While isopropyl alcohol application to the paw pads is described in case studies [15,22] and veterinary reports [23], there is no current empirical evidence recommending the use of isopropyl alcohol for cooling [24]. Furthermore, the use of isopropyl alcohol application to the skin is discouraged based on the possibility of toxicity or the formation of flammable conditions when using defibrillation [25]. Isopropyl alcohol application is an evaporative cooling method, where the process of evaporation transfers thermal energy from the dog’s skin, cooling the dog. However, the cooling recommendation for isopropyl alcohol is to apply it sparingly to paw pads. If isopropyl alcohol is only applied to paw pads, it is unknown if the surface area of the canine paw pad is sufficient to transfer enough heat to adequately decrease core temperature in a dog with exercise-induced hyperthermia. The benefit of isopropyl alcohol in comparison to water immersion is that canine handlers and dog owners in the field could carry a small volume of isopropyl alcohol to facilitate cooling rather than larger jugs of water. If isopropyl alcohol cools at a rate similar to that of water immersion, it may be useful to carry in medical kits for use in environments where pools or large sources of water may not be available for cooling purposes.

Given the wealth of evidence in humans [21,26,27] and some evidence from prior canine studies of cooling methods post-hyperthermia [16,20], it is likely that even partial water immersion would be more effective at cooling dogs post-exertion compared to passive cooling; however, in comparison to full body immersion, partial water immersion has not been extensively examined or compared to passive cooling. Additionally, it is yet unclear how the rate of cooling for water immersion would compare to the rate of cooling for another referenced but unexamined canine cooling method, e.g., the use of isopropyl alcohol on paw pads.

This study examined the relative efficacy of three postexercise cooling methods (partial cool water immersion, isopropyl alcohol on paw pads, and passive cooling) in working dogs and working-dogs-in-training with exertional heat stress with the aim of finding the most effective and efficient way to reduce core body temperature post-exertion. Given the effectiveness of partial water immersion for cooling in humans as compared to passive cooling, it is likely that partial water immersion would also cool at a faster rate than passive cooling in dogs. However, it is yet unclear how the rate of cooling for water immersion would compare to the rate of cooling for another referenced but unexamined canine cooling method, e.g., the use of isopropyl alcohol on paw pads, or how the rate of cooling for isopropyl alcohol application would compare to passive cooling. Temperature and heart rate were examined before and for 20 min after the cooling process to assess the impact of partial water immersion, isopropyl alcohol application to paw pads, and passive cooling on cooling rate and heart rate post-exertion.

## 2. Materials and Methods

### 2.1. Participants

This study enrolled 12 healthy working-dogs-in-training at the Penn Vet Working Dog Center (PVWDC). Inclusion criteria were dogs must be older than 8 months of age, weigh a minimum of 40 pounds (18 kg), be in training and available for the entire study period, have a trained “come/recall” command, and be comfortable with veterinary procedures, restraint, and unfamiliar people. Exclusion criteria included dogs greater than 8 years of age, prior neurologic or musculoskeletal disease, a body condition score of 7–9/9, and aggression or anxiety during veterinary exams. All dogs were evaluated by a veterinarian prior to enrollment and at the start of each study day to ensure compliance with inclusion and exclusion criteria were met. One dog was not present for multiple study days and was dropped as a result, leaving a total of 11 dogs. See Appendix A for further details regarding study participants.

Sample size estimation was carried out in Glimmpse [28] based on a range of estimated standard deviations derived from a similar study [14]. A Hotelling–Lawley Trace with an alpha of 0.05 and a power of 0.8 with scaled standard deviations yields a sample size between 8 and 18. An effect size calculation conducted in G Power [29] based on an alpha of 0.05, power set at 0.8, suggests that the use of 11 participants in a cross-over design can detect a medium effect size (estimated at d = 0.5) with an approximate repeated measurements correlation above 0.65.

### 2.2. Study Design

This study used a randomized cross-over design. Blinding was not possible in this study due to the nature of the interventions. Study participants were split into two groups of six based on the dogs’ training schedules at the PVWDC. Group One had sessions on Mondays and Fridays, while Group Two had sessions on Tuesdays and Thursdays. Dogs participated in three study days over two weeks. Study participants within each group were randomly assigned to one of the three cooling protocols for a particular day using simple randomization with a random number generator (isopropyl alcohol application to the paw pads, cool partial water immersion, or passive cooling/no active intervention); after the study, all dogs completed all three protocols. Dogs participated in a session only once on a given study day, and trials were separated by at least 48 h.

The study protocol was approved by the University of Pennsylvania Institutional Animal Care and Use Committee (IACUC) protocol number 807275.

### 2.3. Technology

Accelerometers and heart rate monitors were placed on each dog, and core temperature-sensing capsules were ingested by each dog at the start of each study day. The activity monitors with an omnidirectional accelerometer (version 3.1, Actical^®^, Respironics, Koninklijke Philips Electronics, Bend, OR, USA) were placed in flat buckle collars that contained a compartment for the accelerometer. The collars were placed on the dog during the physical exam, allowing us to collect quantitative activity data throughout the trial. The accelerometers were calibrated to each individual dog’s weight and age. These monitors have been previously used in exercise studies [7,30,31] and validated in dogs [32]. Accelerometer use was implemented to document if each dog performed a trotting pace during the recall test and to monitor if physical activity such as spinning, barking, or pacing was performed during the cooling period.

Heart rate was continuously monitored throughout exercise and cooling using a human heart rate monitor readily available to the public (Polar USA version H10, Bethpage, NY, USA). A small band with a heart rate monitor was placed on the dog and did not interfere with activity. This monitor measured heart rate every second, allowing for continuous monitoring throughout the study. Heart rate monitor use further validated that the recall test was high intensity and allowed monitoring of post-exercise heart rate recovery to ensure no strenuous activity was being performed that could affect core body temperature. A mobile app (Polar Beat) allowed for continuous heart rate monitoring via mobile devices (Apple iPad Air 4th Generation, Samsung Galaxy Tablet A 2020 SM-T307U, Apple iPhone 7). These heart rate monitors have been used to measure heart rate in dogs in several different studies [33,34,35]. The dogs were shaved on the left ventrolateral aspect of the chest between ribs three and six for monitor placement. Ultrasound gel was applied to the strap, and the monitor was secured to the skin with Johnson & Johnson Elastikon Elastic Tape (4 in) at the time of the physical exam.

Core body temperature was monitored using internal temperature sensing capsules (HQ Inc., Palmetto, FL, USA). Core temperature capsules were administered by mouth using a small amount of wet canned food (Purina Proplan Veterinary Diets EN Gastroenteric Canine Formula). A monitor was assigned to each dog by inputting the specific serial number and calibration number of the ingested pill into the assigned monitor. This monitor allowed for continuous data collection and intermittent readings of core body temperature by scanning the dog’s flank. These capsules have been used in several prior exercise studies [8,9,16]. Capsules were tested to ensure they were functional before administration. A dog was not given a capsule until the previous one was recovered in a bowel movement or if the ingested capsule was unable to be scanned due to a battery malfunction.

The weather was monitored using the National Weather Service website and the Weather Channel App for iPhone on study days. Weather monitoring was based on the Philadelphia Zip Code 19146, where the study was performed. Ambient temperature, humidity percentage, wind speed, and heat index were measured when each individual dog began their sprint test to ensure the conditions were acceptable for outdoor activity and the average heat index was not greater than 93 °F, the maximum set by the research team to ensure safety for participants.

### 2.4. Experimental Protocol

The study was conducted on pre-scheduled days unless there was (1) a heat index of 93 °F (33.89 °C) or more and/or (2) inclement weather that could create a slippery surface or raise safety concerns for the study participants or personnel.

A study day consisted of a physical examination of each dog, warm-up exercises, a recall test, the assigned cooling intervention, and a 20 min cooling period. For data collection, each dog was assigned a consistent handler and data recorder that remained with the dog for the entire study period of each study day. Each handler and data recorder received training prior to the first study day on the study protocol, data collection, data technology, and basic dog-handling skills. One of the study assistants was trained by a licensed veterinarian to recognize behavioral signs of heat stress, exhaustion, and stroke. This study assistant was present during the recall portion of the study to identify signs of heat stress.

#### 2.4.1. Physical Examination

The study day began between 8:15 a.m. and 9:00 a.m. and finished between 10:00 a.m. and 11:00 a.m. EST. Core temperature capsules were administered 30 min to 1 hour prior to the dogs’ physical exams. Water consumption was prohibited from 8 a.m. until the conclusion of the 20 min cooling period so as to not create artifacts for the temperature capsule. Dogs were permitted to drink following the 20 min cool-down period. The physical exam was performed in a temperature-controlled (73 °F, 22.7 °C) indoor environment by a licensed veterinarian. A full physical exam, including axillary temperature, baseline heart rate, baseline respiratory rate, mucus membrane characteristics, capillary refill time, body condition score, hydration status, and cranial skin tent time, was performed. Additionally, a core temperature measurement was recorded and used as a baseline core temperature prior to exercise. At this time, the heart rate monitor and accelerometer collar were placed on the dog, tested to ensure function, and stayed on the dog until the cooling period was complete.

#### 2.4.2. Warm-Up

Following the physical exam, the dog was moved to the recall area for the exertional portion of the test. The recall area was created in a grass fenced-in area by measuring a 25 m runway marked with cones at each end. The handler walked the dog to the recall area, and core body temperature was taken before the warm-up began. The warm-up consisted of stretching exercises based on the PVWDC Fit To Work program (Farr, Ramos, Otto, 2020). Weather information was recorded at this time. See Appendix A for specific weather information per trial day.

#### 2.4.3. Recall Test

The recall test was performed by having the dog run between two recallers 25 m apart for a toy of their choice (a ball, a thin tug, or thick tug pillows). The recall test consisted of the dog being held by the collar by one person (Figure 1). Another person 25 m away with a toy called “[dog’s name], Come”. The dog ran to the person and played tug for a maximum of 15 secs, and the process was repeated in the opposite direction. One of the recallers was always the individual trained to monitor dogs for signs of heat stress (scooped or elongated tongue, ears pulled back, squinting eyes, disengagement, or shade seeking). The other recaller was the dog’s specific handler. Another study assistant timed the recall duration and scanned the dog for a core temperature. A camera on a tripod was used to record the recall test.

Core temperature was recorded before the recall test began (recall 0). After every down and back recall (50 m total), the core temperature was recorded, as well as the heart rate from the Polar Beat app. Proper placement of the monitor on the dog was ensured before the heart rate was recorded. Additional comments were also recorded at this time. These included if the dog became distracted, did not want to run or play, or if a long line needed to be used to keep the dog running in the designated area. Lastly, the dog was checked for signs of heat stress (every 25 m).

The recall portion of the test concluded when one of three scenarios was reached: if the core temperature monitor read 105.0 °F (40.56 °C) or above, if the total recall duration reached 10 min, or if multiple signs of heat stress were demonstrated by the dog. If the dog did not reach 10 min, their total recall duration time was recorded. The signs of heat stress were monitored by the recaller who participated in all recall tests for each dog in the trial. This individual first looked for a scooped or elongated tongue, followed by ears in a back position or squinting eyes, disengagement, or seeking shade. The trial was stopped for heat stress if two or more of these signs were observed.

#### 2.4.4. Cooling Interventions

After completion of the recall, the dog was led to the cooling area in a consistently shaded location under trees, which included wire crates (Midwest iCrate) for resting, a wire crate for the isopropyl alcohol protocol, and two pools. A “rescue area” consisting of one of the pools, towels, and a water bowl was set up in the shaded area in case signs of heat exhaustion were observed, or core body temperature was still elevated above 103 °F (39.4 °C) after the cooling period. Pools were filled using hose water.

The dogs participated in one of three cooling protocols on each day of trials: 30 s of partial water immersion in a 72 °F (22.2 °C) pool, isopropyl alcohol application on the paw pads for 30 s, or no active cooling. Temperature readings were performed immediately after the recalls (timepoint 1). The dogs then received their assigned cooling intervention, which took 30 s to perform. After this intervention, the dog’s temperature was taken again (timepoint 2). Dogs were then led to the wire cooling crate, where their temperature was taken every minute for 20 min (timepoints 3–22). After each of these protocols was performed, the dog was placed in a wire crate and monitored for 20 min. The dog was allowed to sit, lay, or stand. The handler was allowed to provide intermittent kibble as required for enrichment. If the dog’s temperature reached 107.0 °F (41.7 °C) at any time during the study, the dog was immediately placed into the rescue protocol performed by a veterinarian. If the dog’s temperature was still above 103.0 °F (39. 4 °C) after 20 min of cooling, a veterinarian was notified, the dog was required to go to the rescue area, and active cooling was pursued.

The partial water immersion protocol consisted of the dog standing in a Yaheetech Foldable Outdoor Hard Plastic Dog and Cat Swimming pool [36] in shoulder-depth water for 30 s. The pool temperature was monitored with a temperature probe (Traceable WD-20250-01 Type-K/J single-input thermocouple thermometer) approximately every 10 min to ensure it remained at 72.0 °F (22.2 °C). Ice was added as required to maintain the temperature. Specific behavior in the pool, such as standing, laying, sitting, splashing, or attempting to drink the water, was recorded.

The isopropyl alcohol protocol consisted of a wire kennel lined with isopropyl-alcohol-soaked towels that were changed for each dog. A plastic spray bottle was filled with 950 mL of isopropyl alcohol. Three 24” × 16” towels were folded in quarters, and alcohol was poured onto them uniformly on each side. Towels were then laid out flat on the bottom of the crate, side by side, covering the entire floor of the crate. The remaining alcohol in the spray bottle was sprayed on the towels to create a uniform saturation on the towels. Dogs were led into the crate and stood on the towels for 30 s. Behavior during the protocol, such as lifting feet or reluctance to enter the crate, was noted.

The passive cooling protocol consisted of 30 s of standing in the shade before entering the crate to standardize the timing of no active cooling with the other protocols.

Following each cooling method, the dog was led to the wire cooling crate for 20 min.

### 2.5. Statistical Models

All models were generated in R version 4.2.3 using the nlme package and the stats package. One dog was not compliant with the water immersion protocol, and his water immersion data were dropped from the full dataset.

Model 1: Effect of Cooling Protocol and Timepoint on Temperature

A linear mixed-effects model was used to assess the effect of the cooling protocol (no intervention, isopropyl alcohol on paws, or partial water immersion) and time post-exertion on dogs’ core body temperature. A fully specified model included cooling intervention order, breed, and individual dog as random intercepts as well as time post-exertion by a dog as a random slope. To choose the best fit model, Akaike Information Criterion (AIC) values (a measure of prediction error for models) were compared. Lower AIC values indicate higher-quality models. AIC values were compared for the reduced models and the fully specified model. The best-fit model included a random intercept of individual dogs. The alpha value was set at 0.05.

Model 2: Effect of Cooling Protocol and Timepoint on Heart Rate

A linear model was used to assess the effect of the cooling protocol (no intervention, isopropyl alcohol, or partial water immersion) and timepoint post-exertion on dogs’ heart rates. This was carried out as a standard linear model averaged across dogs due to heart rate monitor errors leading to missing data in individual dogs. The alpha value was set at 0.05.

## 3. Results

Over the six trial days, the mean heat index was 83.79 °F ± 7.52 °F (28.77 °C ± 4.18 °C) with a mean humidity of 68.16% ± 10.5% and an ambient temperature of 80.07 °F ± 4.32 °F (26.70 °C ± 2.40 °C). Dogs’ mean running time was 388.85 s ± 137.52 s (6 min and 28.85 s), and the most common reason for stopping the recall test was the display of multiple signs of heat stress. The most common heat stress sign seen was a scooped and elongated tongue. The mean core body temperature the dogs reached at the end of their recall test, calculated using the final temperature recorded for each recall trial performed, was 103.09 °F ± 0.62 °F (39.50 °C ± 0.34 °C).

The accelerometer data were not able to be analyzed due to equipment malfunction errors during data extrapolation from the collars which rendered the data incomplete for analysis. A review of the video taken during the sprint demonstrated all dogs trotted or sprinted during the recall. A video review of the cooling period did not reveal any pacing, persistent barking, or spinning during the cooling period, which would have been captured by the accelerometers.

Model 1: Effect of Cooling Protocol and Timepoint on Temperature

There was a significant main effect of the cooling protocol such that both water immersion (B_1_ = −0.42, t(662) = −4.38, *p* < 0.0001) and isopropyl alcohol (B_1_ = −0.71, t(662) = −7.46, *p* < 0.0001) cooled the dog overall more than passive cooling intervention. Additionally, partial water immersion cooled dogs significantly more than isopropyl alcohol, B_1_ = 0.29, t(662) = 2.95, *p* = 0.003.

There was no main effect of Timepoint, B_1_ = 0.002, t(662) = 0.25, *p* = 0.80.

There is a significant interaction between the cooling protocol and timepoint such that water immersion cooled dogs at a faster rate than passive cooling intervention (B_1_ = 0.03, t(662) = −4.24, *p* < 0.0001) and isopropyl alcohol (B_1_ = 0.29, t(662) = 2.95, *p* = 0.003). Isopropyl alcohol also cooled the dogs faster than passive cooling intervention, B_1_ = 0.70, t(662) = 7.46, *p* < 0.0001. Figure 2 shows a graph of dogs’ temperature over time as it relates to each cooling protocol.

Model 2: Effect of Cooling Protocol and Timepoint on Heart Rate

There is a significant main effect of Timepoint such that heart rate reduces linearly as time increases, t = −22.472, *p* < 0.0001. There is a significant main effect of the Cooling Protocol such that the Isopropyl Alcohol condition had a higher heart rate overall than both Passive Cooling Intervention (t = −10.77, *p* < 0.0001) and Partial Water Immersion (t = −9.27, *p* < 0.001). Passive Cooling Intervention did not have a significantly different overall heart rate compared to Partial Water Immersion (t = 1.50, *p* = 0.13).

There is a significant interaction between Timepoint and Cooling Protocol such that Isopropyl Alcohol and Partial Water Immersion have different rates of heart rate reduction over time (t = −9.12, *p* < 0.0001), as do Partial Water Immersion and Passive Cooling Intervention (t = −9.71, *p* < 0.0001). There is no difference between Isopropyl Alcohol and Passive Cooling Intervention (t = 0.59, *p* = 0.56).

Figure 3 shows a graph of dogs’ heart rate over time as it relates to each cooling protocol.

## 4. Discussion

This study compared methods of core body temperature reduction after exercise-induced hyperthermia in working dogs. Two commonly suggested cooling interventions, water immersion and isopropyl alcohol application to the paw pads, were compared to a control passive cooling intervention. The control protocol, in which dogs passively cooled off with no cooling intervention, was slow and ineffective at cooling the dogs. Both partial water immersion and isopropyl alcohol application to the paw pads cooled dogs at a faster rate than the control protocol; further, partial water immersion cooled dogs faster than isopropyl alcohol application. An analysis of dogs’ heart rates during the period following exercise showed that the isopropyl alcohol condition resulted in significantly higher heart rates overall than partial water immersion or the control protocol, and dogs’ heart rates reduced more slowly over time in the isopropyl alcohol condition than the water immersion or control condition. These results are discussed in further detail below.

Passive cooling does not effectively reduce dogs’ temperature after exercise-induced hyperthermia. Studies in other species have suggested that passive cooling is not as effective as water immersion or evaporative cooling methods, and studies in humans have demonstrated that body temperature can rise after cessation of exercise if an active cooling intervention is not implemented. This study finds a similar result in dogs, in which dogs’ temperature continued to rise after they stopped exercise and began passive cooling, and the passive cooling did not lower dogs’ temperature in comparison to their starting temperature in the 20 min period post-exercise. Dogs’ body temperatures continued to rise after exercise was stopped and the dogs were placed in a wire kennel, reaching on average almost 105 °F (40.7 °C), with a maximum of 106.7 °F (41.5 °C) for one dog. On average, dogs ended the twenty-minute cooling period 0.11 °F (0.06 °C) hotter than they were at the start of the period, with only 2 out of 11 dogs at a lower temperature after 20 min of passive cooling. This suggests that intervention is necessary to reduce dogs’ body temperature post-exercise and to reduce their risk of heat stress and heat stroke.

Dogs in the water immersion condition had an overall lower average temperature in the 20 min post-exercise and post-cooling-intervention than dogs in the control protocol who had no cooling intervention and dogs in the isopropyl alcohol application condition. Further, partial water immersion cooled dogs at a significantly faster *rate* than both isopropyl alcohol application to paws and passive cooling intervention. This result provides more evidence for the use of water immersion as a first-line of defense if cooling is required for working dogs. Researchers and practitioners have long advocated for the use of water immersion as a cooling mechanism, with findings from human and canine literature recommending water immersion as the gold standard cooling technique for post-exertional hyperthermia [13,18,19]. Further research using similar conductive cooling methods involving partial water immersion or continuous water application has been conducted in humans [21,37,38] and horses [39,40], and these methods have also been shown to be successful at cooling post-exercise-induced hyperthermia.

However, there is disagreement in the literature regarding the ideal temperature for water immersion cooling. One canine study exploring the effect of different temperatures of water in a water immersion protocol found equal effectiveness of water immersion across temperatures [20]. These researchers suggested that the use of ice water (close to 32 °F or 0 °C) could have caused shivering and vasoconstriction in the skin, leading to more heat retention and the same cooling rate as cool or room-temperature water despite the larger temperature gradient. Other veterinary researchers suggest that cold water (33.8 to 51.8 °F; 1 to 11 °C) immersion is the most effective [41], as cooling too slowly has been shown to increase the risk of heat-related illness severity in humans [42]. In humans, there have been mixed results, in which some studies find that ice water (35.6 °F; 2 °C) has provided a faster rate of cooling than cool (46.4 °F; 8 °C) or room-temperature water (68 °F; 20 °C) [43], and others find no effect of temperature [17]. This study used room temperature (72 °F; 22.2 °C) water in a pool, which may be more accessible for use than colder water that may require ice or insulation to remain cold in a field setting, and we found that room temperature water was effective in reducing dogs’ body temperature.

In addition to partial water immersion, isopropyl alcohol application to paw pads reduced dogs’ core body temperature more than passive cooling intervention; however, isopropyl alcohol lowered temperature more slowly than water. While isopropyl alcohol application has previously been suggested and utilized as a method of cooling [15,24], there were no existing studies demonstrating its efficacy for exercise-induced hyperthermia. Studies in veterinary surgery examining the effects of different preoperatory rinses on dog and cat body temperature found that isopropyl alcohol rinses lead to greater cooling than water rinses, and some found no difference between isopropyl alcohol and water [44]. This is the first study to demonstrate that brief isopropyl alcohol application can accelerate the cooling process in dogs with exercise-induced hyperthermia, and additionally shows that isopropyl alcohol application to the paw pads is not as effective of a cooling method as partial water immersion.

However, isopropyl alcohol application significantly *increased* dogs’ heart rates in comparison to the other conditions. One potential reason this intervention increased dogs’ heart rate is that isopropyl alcohol is an irritant [45]. Breathing in its aversive odor [46] has been known to increase animals’ heart rates. The odor and discomfort caused by contact with isopropyl alcohol could cause stress, which is also known to raise heart rate [45,47]. As such, isopropyl alcohol, while more effective than no cooling intervention at all at lowering the temperature, can counterproductively cause stress and increase heart rate, which in turn could raise the dogs’ temperature.

The isopropyl alcohol in this intervention was applied only to the paw pads and only for 30 s; other studies have suggested that isopropyl alcohol application over a larger area would cool dogs faster. However, one important factor in a cooling intervention is the safety of use. The intervention must not cause undue harm during use and must be acceptable for potential frequent uses or preventative uses, given that working dogs often spend long hours outdoors in the heat. Isopropyl alcohol application can cause skin irritation if applied too frequently and pain if accidentally applied on an abrasion. The continued use of isopropyl alcohol on the paw pads could cause them to crack and become damaged. This could result in injury to the pad, which could limit a dog’s ability to work. Isopropyl alcohol is also highly flammable and must be minimally applied to reduce the risk of fire or explosion. While the larger application of isopropyl alcohol on the dogs’ bodies may cool them faster, it is not worth the risk and stress it may cause the dog.

One benefit of isopropyl alcohol in comparison to water immersion is its portability for field cooling. Water immersion requires a pool and enough water to immerse the dog. For working dogs, this could be more water than can be reasonably carried in the field. Isopropyl alcohol can be easily kept in a medical kit for emergencies. We recognize that in the field, supplies may be limited. The isopropyl alcohol did have an effect on core body temperature and can be used when water is not available since its effects were more helpful in reducing core body temperature when compared to no active cooling. It should be kept in first aid kits but not relied on as a first-line cooling method due to its potentially harmful effects. More portable methods of cooling must be explored to replace isopropyl alcohol.

### Limitations

One natural limitation of our protocol is that the volume of water that got onto the dog during the water immersion protocol was dependent on the behavior of the dog in the pool. The majority of dogs stood in the pool and would not lie down. This made it difficult for water to reach core areas, such as the chest. However, this behavior still produced decreasing core body temperatures for most participants and was still the most effective protocol. Other methods of water immersion, including a whole-body shower/hose application, may produce an increase in cooling effect; however, these methods are difficult to use in a field environment. If this study were to be repeated, participants could be trained to lay in the pool, or a hose could be used, which may result in more dramatic cooling.

A second limitation was inadvertent water consumption during the experiment. This protocol purposely limited dogs’ consumption of water during the cooling protocol due to the potential impact on temperature pill readings. Regardless, some of the dogs managed to drink a small amount of the pool water after exercise, even though their heads were encouraged to remain out of the water by their handler. This raised concerns regarding the accuracy of the core temperature monitors when they came in contact with the cool water. In a study examining the effect of cool water consumption on temperature pill readings in firefighters, the ingestion of cool water temporarily lowered the temperature reading from the pills. As the time after water consumption increased, the temporary decrement in temperature became negligible [48]. Our study aimed to increase the amount of time between ingestion of the core temperature monitor and the possible water ingestion to avoid this disruption. The one dog that consumed a great deal of water during the immersion protocol had his water immersion data dropped due to a lack of compliance and the potential for incorrect data, as described above in the statistical model’s section.

The isopropyl alcohol protocol was an anticipated obstacle for the study. It was difficult to determine a consistent way to quantify a specific amount of alcohol applied to the paw pads for each dog. The procedure of folding the towels and spraying a fixed amount of alcohol on the towels proved to be the best method. However, the dogs were hesitant to walk over an isopropyl alcohol-soaked towel placed on a yoga mat due to the odor. By having the towels in the crate ahead of time, allowing the odor to dissipate before use, and luring with treats, we eliminated some of the hesitancy and further standardized the procedure between dogs.

Lastly, the duration of the cooling intervention was shorter than the generally recommended cooling intervention. The dogs in this study stood in pool water for 30 s. In contrast, one study examining methods of cooling in working dogs used a 5 min immersion period [16]. They found water immersion to be more effective than passive cooling despite the duration difference [16]. In humans, many different immersion durations have been used, from 3 min to 60 min; however, a meta-analysis found that under 10 min of immersion was more effective than longer immersion durations [27]. Despite the shorter duration of water immersion, the dogs showed significant cooling, and the intervention was more successful than passive cooling or isopropyl alcohol application to paw pads.

## 5. Conclusions

The purpose of this study was to investigate possible solutions for a common consequence of intense physical activity in high-temperature environments: heat stress with the possibility of progressing to heat stroke. The current literature does not provide a recommendation for the most effective way to reduce a dog’s core body temperature following exercise-induced hyperthermia. This information is vital to allow working dogs to continue to perform in conditions of increased ambient temperature and to prevent the consequences of hyperthermia-induced injuries.

This study examined one conductive and one evaporative cooling method, partial water immersion and isopropyl alcohol application to paw pads, respectively, which are currently utilized in the field for working dogs. We found that a partial water immersion protocol was more effective than isopropyl alcohol application to the paw pads and passive cooling in reducing the core body temperature of working dogs after post-exercise-induced hyperthermia. When both water and isopropyl alcohol are available, water immersion should be the preferred method of cooling due to its higher effectiveness, tolerance of use, and safety.

The conclusions of this study apply to working dogs but also to those of similar breed and activity levels. Many pet dogs present to emergency veterinarians for heat-induced clinical signs, and those whose owners cooled them prior to admission had a more favorable outcome [12]. If owners can apply our findings as well as recognize common signs of heat stress, heat-injury-related deaths may be avoided.

Future studies should examine the applications of the water immersion concept to field use where a pool is not readily available. This could include examining pre-cooling techniques, the use of cooling collars, or water application to specific areas of the body.

## Figures and Tables

**Figure 1 animals-13-03673-f001:**
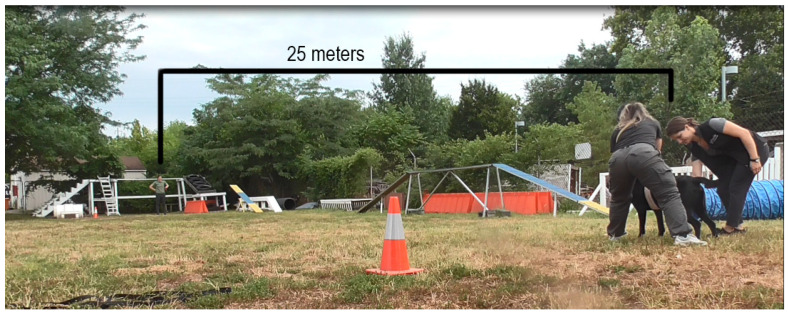
Set up of recall test. The dog is wearing a heart rate monitor on a band around their torso. The handler releases the dog to sprint back and forth between them and a secondary handler. Another person collects the dog’s temperature and monitors for signs of heat stress.

**Figure 2 animals-13-03673-f002:**
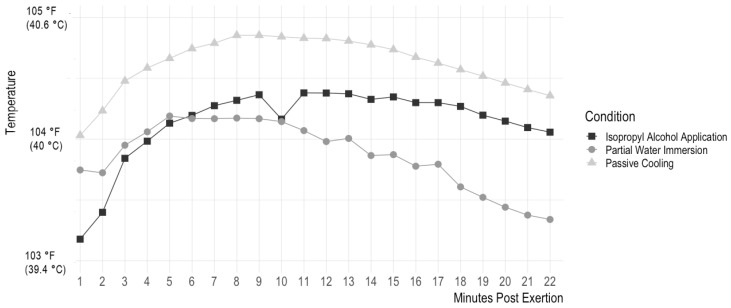
Effect of cooling intervention on core body temperature over time, beginning post-exertion. This graph shows the effect of the three different cooling interventions (isopropyl alcohol application, passive cooling, and partial water immersion) on dogs’ temperature in Fahrenheit over time. This graph begins at the point after dogs completed recalls (Minute 1 post-exertion). The dogs then received a 30-s cooling intervention, and their temperature was measured again (Minute 2 post-exertion). The dogs completed their assigned cooling intervention and were placed in a wire cooling crate, where their temperature was measured every minute for 20 min.

**Figure 3 animals-13-03673-f003:**
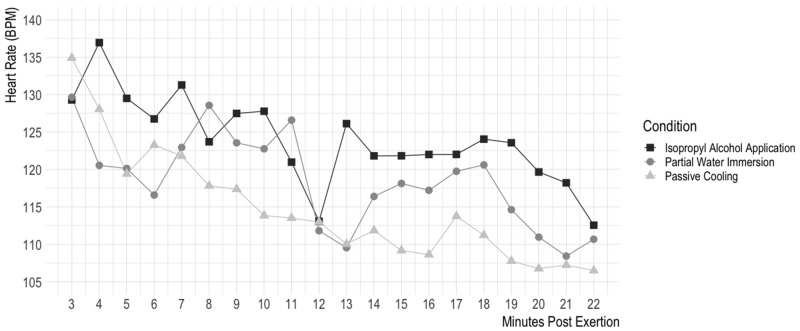
Effect of cooling intervention on heart rate over time, beginning post-cooling intervention. This graph shows the effect of the three different cooling interventions (isopropyl alcohol application, passive cooling, and partial water immersion) on dogs’ heart rates in beats per minute over time. Due to an error in the heart rate monitors, heart rate monitoring began when the dogs were placed in the wire cooling crate for 20 min rather than immediately post-exertion (this point in time is 3 min post-exertion). As such, heart rate was tracked for 20 min beginning 3 min post-exertion rather than the 22 min of temperature tracking beginning immediately after post-exertion seen in Figure 2. In the wire cooling crate, heart rate was measured every second for 20 min; for visual clarity, this graph shows heart rate averaged over 1 min intervals. Statistical analyses were carried out using the full set of data.

## Data Availability

The data presented in this study are available on request from the corresponding author.

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
