# Peer review of "A Randomized Cross-Over Study Comparing Cooling Methods for Exercise-Induced Hyperthermia in Working Dogs in Training"

_animals, 2023, doi:10.3390/ani13233673_

Round 1
Reviewer 1 Report
Comments and Suggestions for Authors
Thank you for the opportunity to review this very interesting manuscript which is targeting an important topic in veterinary medicine, canine sports and performance medicine and canine welfare. Exertional hyperthermia in dogs is a continuum of challenging conditions and a highly relevant topic for clinicians, stakeholders (such as dog owners, organizers of competitions and trials) and researchers.
General comments on this version of the manuscript: The introduction is solid and the results are fairly clearly presented. However, there are results missing with regards to the outcome measures presented in the methods section. In the material and methods section there is a lack of information to the reader, since the Supplementary Table is missing.
Title: Comparing Cooling Methods for Exercise-Induced Exertional Hyperthermia in Working Dogs.
The use of “exercise-induced” as well as “exertional” to describe the background to hyperthermia is superfluous. In my opinion one of the expressions are to be used. Please revise accordingly. Also, consider adding a description of the study design to the title as well as reconsider the description of the study sample.
Introduction
To increase the readability, make sure that all numbers that refer to temperatures are presented in Fahrenheit AND Celsius. Please change accordingly throughout the whole manuscript.
Line 79-82 This sentence doesn’t belong in the introduction.
There is a recent and relevant reference that might be interesting to the authors of this manuscript. Hall et al. Cooling Methods Used to Manage Heat-Related Illness in Dogs Presented to Primary Care Veterinary Practices during 2016–2018 in the UK.
Line 109-111 Please clarify the aim of this study. It is pretty straightforward in reality, but what is relative efficacy in this study here? I think efficacy is what the authors are aiming to assess. But there is nothing in the aim about what outcome the authors are targeting – core body temperature. Heart rate?
There are three measurement methods used? And the authors hypothesize about the time it takes for the dog to go back to baseline core temperature only.
Materials and methods
Under this sub-heading I miss a section about the study design. Please add a clear description of the study design used. There is some information found in Line 127 and Line 130.
Unfortunately, I can’t find the Supplementary Table 1 with the details of the study participants. Please add.
Line 126 To distinguish from the tables in the manuscript, please change from “Supplementary Table 1” to “Supplementary Table S1”
Line 140-149 This is a description of measurement used during the trial and can’t see that the authors present any of the results from the accelerometer in the results. Please correct this throughout the whole manuscript. Why is it relevant to use the accelerometer in this study of exertional hyperthermia?
Line 151-153 Which type of Polar heart rate monitor was used? With regards to the high intensity activity measured, it is relevant to inform the reader about the sampling frequency of the device.
The authors refer to a study assessing heart rate in dogs walking on a treadmill and the feasibility of heart rate measurements. There are however other studies on the validity and reliability of the Polar heart rate monitor in dogs. Please add appropriate references. In addition, the authors will be able to elaborate the finding in this study with regards to repeated measurements and the known measurement error of the Polar device. And, why is it relevant to use the heart rate measurement in this study of exertional hyperthermia?
Line 185 For how long time was the heart rate monitor used during the 20 minutes cooling period? Figure 2 (or 3?) illustrates 12 minutes, while Figure 1 (or 2) illustrates temperature over a longer time period?
Line 225 The signs of heat-stress is well worth mentioning. But, the same phrasing is repeated three times under the same section. 2.3.3 Recall. See Line 240-242 and Line 248-249. Please revise to avoid unnecessary repetitions.
Line 307-308 Maybe there is a typo here. I assume the authors mean “heart rate in beats per minute” instead of “core body temperature”?
Results
Again. Please make sure all numbers that refer to core body temperature are presented both in F and C. Did the authors dropp the measurements from the accelerometer?
Line 315 “Average” may refer to mean or median. I can see that the authors used mean +/- SD, but please change the wording to what you used.
Line 317 I somehow miss the mean+/-SD or equivalent from the heart rate monitoring and the accelerometer.
Line 329 Figure 1 is the set-up of the recall. Please revise the manuscript throughout and change to correct number to the figures.
Figure captions page 8 and 9 Please move the title of the figure to the figure caption. Change to -
“Figure 1. Effect of cooling intervention on core body temperature over time. This graph….”
I would love to see an illustration of the Celcius degrees over time as well.
Discussion
Line 357 The participants were dogs older than 8 months, > 40 pounds and of various breeds. Is it relevant that the dogs were in training to become working dogs? What specific characteristics do the dogs in this sample have that make them different from other dogs that don’t undergo training to become working dogs? Please consider this since your answers to these questions may influence the external validity and to what extent the results here can be generalized or extended to dogs, settings and characteristics in other than those included in this particular demonstration.
I would strongly consider to erase “working” from the title and throughout the whole manuscript, since the authors have a good description of the study group. I assume that other dogs that fit to the description, and not only those in training to become working dogs, are able to gallop according to the recall test.
Line 375-381 This is such an interesting discussion! Good job.
Conclusion
This conclusion is not clear. Line 491-500 is not relevant for the conclusion – move this part to the discussion? The conclusion goes back to the aim of this study. When the authors work on the aim, I think that part of the conclusion will match really good.
I asked before about the external validity of this study. Are the results from this study applicable in other dogs similar to those in the study sample? What if the other dogs are not dogs in training to become working dogs? What I am going for is… if the results are applicable to other similar dogs, not only specifically dogs in training to become certified working dogs, the generalizability and external validity of this study increases. I think the results are possible to transfer to other dogs as well. Of course not brachy dogs and such, but dogs with the same characteristics, not only the same working goal.
References
Line 574 Is there something missing here?
Line 585 This is not a methodological study of validity and reliability of the Polar heart rate monitor used in this study. Validity and reliability the Polar heart rate monitor in dogs has been reported elsewhere though. Please change or at least add a proper reference.
Veterinary Emergency and Critical Care Society (2016) DOI:10.1111/vec.12455 includes water immersion in the pre-hospital treatment protocol.
A recent paper from the field: Hall et al 2023 - Cooling Methods Used to Manage Heat-Related Illness in Dogs Presented to Primary Care Veterinary Practices during 2016–2018 in the UK.
Author Response
Reviewer 1:
- Thank you for the opportunity to review this very interesting manuscript which is targeting an important topic in veterinary medicine, canine sports and performance medicine and canine welfare. Exertional hyperthermia in dogs is a continuum of challenging conditions and a highly relevant topic for clinicians, stakeholders (such as dog owners, organizers of competitions and trials) and researchers. General comments on this version of the manuscript: The introduction is solid and the results are fairly clearly presented. However, there are results missing with regards to the outcome measures presented in the methods section.
Thank you for your detailed and informative review. Below each of your comments is a clarification, response, or documentation of the changes we made based on your review.
- In the material and methods section there is a lack of information to the reader, since the Supplementary Table is missing.
We have included a supplementary table, our apologies that it was not shared during the initial review.
- Title: Comparing Cooling Methods for Exercise-Induced Exertional Hyperthermia in Working Dogs. The use of “exercise-induced” as well as “exertional” to describe the background to hyperthermia is superfluous. In my opinion one of the expressions are to be used. Please revise accordingly. Also, consider adding a description of the study design to the title as well as reconsider the description of the study sample.
The title to “A Randomized Cross-Over Study Comparing Cooling Methods (Partial Water Immersion versus Isopropyl Alcohol Application to the Paw Pads) for Exercise-Induced Hyperthermia in Working Dogs in Training”
Introduction
- To increase the readability, make sure that all numbers that refer to temperatures are presented in Fahrenheit AND Celsius. Please change accordingly throughout the whole manuscript.
The correction has been made throughout the manuscript.
- Line 79-82 This sentence doesn’t belong in the introduction.
The sentence has been moved from the introduction to the conclusion.
- There is a recent and relevant reference that might be interesting to the authors of this manuscript. Hall et al. Cooling Methods Used to Manage Heat-Related Illness in Dogs Presented to Primary Care Veterinary Practices during 2016–2018 in the UK.
Thank you, we have added text about this article to the introduction to further strengthen the evidence against use of isopropyl alcohol for cooling, especially in the emergency setting.
“While isopropyl alcohol application to the paw pads is described in case studies ​[20,21]​ and veterinary reports ​[22]​ there is no current empirical evidence recommending the use of isopropyl alcohol for cooling ​[23]​. Furthermore, the use of isopropyl alcohol application to the skin is discouraged based on the possibility of toxicity or the formation of flammable conditions when using defibrillation ​[24]​. “
- Line 109-111 Please clarify the aim of this study. It is pretty straightforward in reality, but what is relative efficacy in this study here? I think efficacy is what the authors are aiming to assess. But there is nothing in the aim about what outcome the authors are targeting – core body temperature. Heart rate?
Clarification has been added regarding the aim of the study to lines.
“As such, this study examined the relative efficacy of three postexercise cooling methods (partial cool water immersion, isopropyl alcohol on paw pads, and passive cooling) in working dogs and working-dogs-in-training with exertional heat stress with the aim of finding the most effective and efficient way to reduce core body temperature post-exertion. “
- There are three measurement methods used? And the authors hypothesize about the time it takes for the dog to go back to baseline core temperature only.
Core temperature and heart rate were monitored and analyzed for this study. We did obtain accelerometer data however the data recovery was inhibited by equipment malfunctions during data recovery and thus unable to be analyzed. The text below has been added to the results section.
“The accelerometer data were not able to be analyzed due to equipment malfunction errors in data extrapolation which rendered the data incomplete for analysis. Review of video taken during the sprint demonstrated all dogs trotted or sprinted during the recall. “
- Materials and methods - Under this sub-heading I miss a section about the study design. Please add a clear description of the study design used. There is some information found in Line 127 and Line 130.
A subheading called study design and included the study type.
“2.2. Study Design
This study used a randomized crossover design. Blinding was not possible in this study due to the nature of the interventions.”
- Unfortunately, I can’t find the Supplementary Table 1 with the details of the study participants. Please add.
The supplementary table was emailed to the editor and reviewer. If reviewers are still unable to access the table please let us know.
- Line 126 To distinguish from the tables in the manuscript, please change from “Supplementary Table 1” to “Supplementary Table S1”
Thank you for your suggestion. The suggested changes have been implemented.
- Line 140-149 This is a description of measurement used during the trial and can’t see that the authors present any of the results from the accelerometer in the results. Please correct this throughout the whole manuscript. Why is it relevant to use the accelerometer in this study of exertional hyperthermia?
We did obtain accelerometer data however the data recovery was inhibited by equipment malfunctions during data recovery and thus unable to be analyzed. This is reflected in the results section.
“The accelerometers data was not able to be analyzed due to equipment malfunction errors in data extrapolation which rendered the data incomplete for analysis. Review of video taken during the sprint demonstrated all dogs trotted or sprinted during the recall. “
- Line 151-153 Which type of Polar heart rate monitor was used? With regards to the high intensity activity measured, it is relevant to inform the reader about the sampling frequency of the device. The authors refer to a study assessing heart rate in dogs walking on a treadmill and the feasibility of heart rate measurements. There are however other studies on the validity and reliability of the Polar heart rate monitor in dogs. Please add appropriate references. In addition, the authors will be able to elaborate the finding in this study with regards to repeated measurements and the known measurement error of the Polar device. And, why is it relevant to use the heart rate measurement in this study of exertional hyperthermia?
The model of polar heart rate used (H10) and sampling frequency have been added. We have included the reasoning as to why we sampled heart rate during the cooling period. The heart rate was used as an objective measure to monitor physical activity and stress during the cooling period that may have impacted body temperature.
“Heart rate was continuously monitored throughout exercise and cooling using a human heart rate monitor readily available to the public (Polar USA version H10, Bethpage, NY, USA). A small band with the heart rate monitor was placed on the dog and did not interfere with activity. This monitor measured heart rate every second, allowing for continuous monitoring throughout the study. Heart rate monitor use further validated that the recall test was high intensity and allowed monitoring of post exercise heart rate recovery to ensure no strenuous activity was being performed that could affect core body temperature. A mobile app (Polar Beat) allowed for continuous heart rate monitoring via mobile devices (Apple IPad Air 4th Generation, Samsung Galaxy Tablet A 2020 SM-T307U, Apple IPhone 7). These heart rate monitors have been used to measure heart rate in dogs in several different studies ​[33–35]. Dogs ​were shaved on the left ventrolateral aspect of the chest between ribs three and six for monitor placement. Ultrasound gel was applied to the strap and the monitor was secured to the skin with Johnson & Johnson Elastikon Elastic Tape (4 in) at the time of the physical exam (see Figure 1 for heart rate strap location).  “
- Line 185 For how long time was the heart rate monitor used during the 20 minutes cooling period? Figure 2 (or 3?) illustrates 12 minutes, while Figure 1 (or 2) illustrates temperature over a longer time period?
Thank you, the heart rate graph has been edited. The previous graph showed temperature averaged over approximately 2-min periods (leading to about 12 time points). The current graph shows temperature averaged over 1-min periods (20 time points). Temperature was measured for 2 more minutes than heart rate was measured due to heart rate monitor equipment error - the heart rate collection began two minutes later than the temperature collection. Since the heart rate and temperature are not included in the same analysis, the full collected data for both heart rate and temperature were used in their respective analyses, even though temperature was collected for 2 additional minutes.
- Line 225 The signs of heat-stress are well worth mentioning. But, the same phrasing is repeated three times under the same section. 2.3.3 Recall. See Line 240-242 and Line 248-249. Please revise to avoid unnecessary repetitions.
Thank you, the heat stress signals were removed in lines 240-242.
- Line 307-308 Maybe there is a typo here. I assume the authors mean “heart rate in beats per minute” instead of “core body temperature”?
Thank you for identifying this typo, the change has been made.
- Results - Again. Please make sure all numbers that refer to core body temperature are presented both in F and C. Did the authors dropp the measurements from the accelerometer?
Changes were implemented. Further discussion of accelerometer data is within comment #12.
- Line 315 “Average” may refer to mean or median. I can see that the authors used mean +/- SD, but please change the wording to what you used.
We have changed the word average to mean throughout.
- Line 317 I somehow miss the mean+/-SD or equivalent from the heart rate monitoring and the accelerometer.
We have added in the heart rate monitor mean and SD.
- Line 329 Figure 1 is the set-up of the recall. Please revise the manuscript throughout and change to correct number to the figures.
Thank you for catching this error. We have corrected the figure numbers in the manuscript.
- Figure captions page 8 and 9 Please move the title of the figure to the figure caption. Change to -“Figure 1. Effect of cooling intervention on core body temperature over time. This graph….”
Thank you, we have made this change and edited the figure numbers.
- I would love to see an illustration of the Celcius degrees over time as well.
Celsius conversions have been added to the existing graph.
- Discussion- Line 357 The participants were dogs older than 8 months, > 40 pounds and of various breeds. Is it relevant that the dogs were in training to become working dogs? What specific characteristics do the dogs in this sample have that make them different from other dogs that don’t undergo training to become working dogs? Please consider this since your answers to these questions may influence the external validity and to what extent the results here can be generalized or extended to dogs, settings and characteristics in other than those included in this particular demonstration. I would strongly consider to erase “working” from the title and throughout the whole manuscript, since the authors have a good description of the study group. I assume that other dogs that fit to the description, and not only those in training to become working dogs, are able to gallop according to the recall test.
Thank you for your suggestion and input on this topic. The relevance of training here is that the working dogs in training regularly work on fitness, are within a healthy body score range, and are high drive and well-conditioned. This would not extrapolate to all dogs, some of which would be less conditioned, lower drive (relevant to the performance/commitment given to the sprint test exercise portion) and may be over- or underweight, which would impact their susceptibility to heat stress and the potential impact of cooling interventions.
You are correct in that the data can be extrapolated to dogs of similar size, conditioning, and body score who are not working dogs. We hope that this data would be helpful for those dogs (for example, similar-sized dogs who do dog sports). However, we feel the distinction of working dogs versus all dogs is important for this manuscript.
- Line 375-381 This is such an interesting discussion! Good job.
Thank you very much.
- Conclusion- This conclusion is not clear. Line 491-500 is not relevant for the conclusion – move this part to the discussion? The conclusion goes back to the aim of this study. When the authors work on the aim, I think that part of the conclusion will match really good.
Thank you for the suggestion. We moved these lines to the discussion section, and we addressed and clarified the aim which was then reflected in the conclusion.
- I asked before about the external validity of this study. Are the results from this study applicable in other dogs similar to those in the study sample? What if the other dogs are not dogs in training to become working dogs? What I am going for is… if the results are applicable to other similar dogs, not only specifically dogs in training to become certified working dogs, the generalizability and external validity of this study increases. I think the results are possible to transfer to other dogs as well. Of course not brachy dogs and such, but dogs with the same characteristics, not only the same working goal.
As commented above we feel the distinction that this was performed in working dogs is important to include in the study, even with some potential generalizability to other dogs of the same size, body condition score, and fitness level. We added a section in the conclusion to include relevance to pet dogs.
“The conclusions from this study are applicable to working dogs but also those of similar breed and activity level. Many pet dogs present to emergency veterinarians for heat induced clinical signs and those whose owners cooled them prior to admission, had a more favorable outcome ​[12]​. If owners can apply our findings as well as recognize common signs of heat stress, heat-injury-related deaths may be avoided.”
References
- Line 574 Is there something missing here?
We have gone through the references and made sure they were complete.
- Line 585 This is not a methodological study of validity and reliability of the Polar heart rate monitor used in this study. Validity and reliability the Polar heart rate monitor in dogs has been reported elsewhere though. Please change or at least add a proper reference.
Several sources have been added for the validity and reliability of polar heart rate monitors.
Reviewer 2 Report
Comments and Suggestions for Authors
The article is very interesting for the veterinarians and operators, that act within the working dog discipline. The project is well done and deserves publication.
I only have a few minor revisions to suggest:
Pag. 3 “partecipants”: I would have appreciated reading data such as breed, sex and age of the dogs used for the study. I imagine they are reported in the supplementary files, to which unfortunately no direct access has been found.
pag 3 line 130: “---randomly assigned1….”. What is the meaning of the superscript number 1? I don't understand what it refers to, as I don't read any legend in the text. There are other similar situations.
pag 3 line 136: I personally prefer the term “handlers” instead of “fosters”.
Pag 4 lines 156-161: for heart rate monitors, authors report the use of Polar USA. Were the dogs wearing a harness to support the device? Could the harness alter the final results? If so, perhaps it should be brought back within limits
Pag 6 lines 248-250. In my opinion, the evauated sign of stress have been excessively repeated in the text. it would be enough to report it only once.
Pag 7 line 302: What is AIC?
Pag 7: Statistical analysis paragraph: No significance level has been reported.
Pag 7 lines 311-312: also data in °C could useful for readers and only in °F
Pag 8 lines 345-346: “..as do Partial Water Immersion and Passive Cooling Intervention (t = 1.50, p = 0.13)”. It seems not significant.
Pag 9 lines 377-378: what is the meaning of “(__C)”
Pag 10 lines 403: attention to double brackets
Pag 10 line 418: why is “increased” in italics
Author Response
The article is very interesting for the veterinarians and operators, that act within the working dog discipline. The project is well done and deserves publication.
I only have a few minor revisions to suggest:
- 3 “partecipants”: I would have appreciated reading data such as breed, sex and age of the dogs used for the study. I imagine they are reported in the supplementary files, to which unfortunately no direct access has been found.
Yes, we have reported this information in supplementary tables. Please let us know if you still cannot access it.
- pag 3 line 130: “---randomly assigned1….”. What is the meaning of the superscript number 1? I don't understand what it refers to, as I don't read any legend in the text. There are other similar situations.
These have been removed, thank you.
- pag 3 line 136: I personally prefer the term “handlers” instead of “fosters”.
Thank you for your input. In this case, all of our dogs are in training, and have not been assigned to a permanent handler. This line has also been removed, because informed consent was not needed; the dogs are all owned by the Penn Vet Working Dog Center.
- Pag 4 lines 156-161: for heart rate monitors, authors report the use of Polar USA. Were the dogs wearing a harness to support the device? Could the harness alter the final results? If so, perhaps it should be brought back within limits
Thank you for your comment. The dogs had the polar monitor within the strap that comes with the device. The strap was secured with elastikon and vet wrap.
“Heart rate was continuously monitored throughout exercise and cooling using a human heart rate monitor readily available to the public (Polar USA version H10, Bethpage, NY, USA). A small band with the heart rate monitor was placed on the dog and did not interfere with activity. This monitor measured heart rate every second, allowing for continuous monitoring throughout the study. Heart rate monitor use further validated that the recall test was high intensity and allowed monitoring of post exercise heart rate recovery to ensure no strenuous activity was being performed that could affect core body temperature. A mobile app (Polar Beat) allowed for continuous heart rate monitoring via mobile devices (Apple IPad Air 4th Generation, Samsung Galaxy Tablet A 2020 SM-T307U, Apple IPhone 7). These heart rate monitors have been used to measure heart rate in dogs in several different studies ​[33–35]. Dogs ​were shaved on the left ventrolateral aspect of the chest between ribs three and six for monitor placement. Ultrasound gel was applied to the strap and the monitor was secured to the skin with Johnson & Johnson Elastikon Elastic Tape (4 in) at the time of the physical exam (see Figure 1 for heart rate strap location).  “
- Pag 6 lines 248-250. In my opinion, the evauated sign of stress have been excessively repeated in the text. it would be enough to report it only once.
Thank you for your suggestion. The change has been made.
- Pag 7 line 302: What is AIC?
AIC is the Akaike Information Criterion, which is a value often used as a model selection tool to assess goodness-of-fit. This information has been added to the manuscript.
- Pag 7: Statistical analysis paragraph: No significance level has been reported.
The alpha value for all analyses was set to 0.05. This was added to the information about statistical models throughout section 2.5.
- Pag 7 lines 311-312: also data in °C could useful for readers and only in °F
Thank you for your suggestion. The change that has been made throughout the manuscript.
- Pag 8 lines 345-346: “..as do Partial Water Immersion and Passive Cooling Intervention (t = 1.50, p = 0.13)”. It seems not significant.
Thank you for catching this, the interaction statistics were accidentally repeated from the main effects statistics. The partial water immersion and passive cooling intervention are significantly different, t = -9.12, p < 0.0001.
- Pag 9 lines 377-378: what is the meaning of “(__C)”
This is a place where we forgot to add the Celsius conversion for a Fahrenheit value. It is added now.
- Pag 10 lines 403: attention to double brackets
Thank you for your suggestion. The change has been implemented.
- Pag 10 line 418: why is “increased” in italics
This is in italics for emphasis.
Reviewer 3 Report
Comments and Suggestions for Authors
Dear authors, I have carefully reviewed the current manuscript. The objective of the study was to assess the efficacy of 3 post exercise cooling methods. It seems an intresting study in a topic that even now we have no clear indications as for the first step proposed strategies for the decrease of the body temperature or for the proper cooling temperatures that should be used (at least in case of a heatstroke). Although someone may think that the current research questions could be easily examined utilizing not such a sophisticated or demanding study design, I have defined several issues that need to be addressed or explained and refer to the current study design.
Introduction section
Line 92: fix the marks/units of Celsious degrees (not only here, but throughout the whole manuscript)
Line 107-108: use a reference for such an emphatic statement
Lines 112-115: "Predict"? This is not a proper way of presenting the hypothesis statement. Can you predict the outcome of the study? Here should be the hypothesis of the study, which should be a testable statement, and not a definite one.
Material and Methods section
The major problem of this section is the many missing (not mentioned) ARRIVE guidelines criteria.
1. Which were the primary outcomes of the study? The average time that the body temperature returned to normal? I assume that the Heart Rate differences were the secondary outcomes of the study? They should be pre-specified and clearly mentioned.
2. What was the type of the study? Was it a pilot or a prospective study?
3. There is no mention of the sample size calculation.
4. What about inclusion/exclusion criteria? You do mention some inclusion criteria, but what about the exclusion criteria? Could you include a 15 years-old dog, for example?
5. You mention that this was a cross-over study (this is only mentioned in the Abstract). Which was the randomization plan? You do mention that the dogs were randomly assigned to each treatment (Lines 130-131), but which was the way of doing that?
6. I assume that there was no blinding throughout the procedure. However, that should have been mentioned, as well.
Results section
Which are the demographic characteristics of the dogs that participated in the study?
Lines 315-316: What is this temperature mentioned reffering to? This is mean and standard deviation of all dogs in the different study days and in the different treatments applied?
Figure 1: What information does this figure provide? Timepoints 1-22 correspond to 1-22 minutes? When did the dog begin to exercise throughout this graph? At which timepoint? And when did the exercise stop? And at which timepoint was the 30second treatment applied?
Discussion section
Lines 386-390: You should mention that the study in horses does not utilize a water immersion. Mention the different strategies (sponging and pouring cold water) of either 9 or 31 degrees of tap water. Apart from the use of cooling (or not) water, there is no other resemblance to the current study design. It is true that it seems a safe first line defence, but as long as you review the literature, you should scrutinize the different approaches utilized.
Lines 393-395: I agree that this is the result of Ref 19, but more recent studies (Flournoy et al. 2003) propose that low temperatures (ice water baths) are contraindicated.
Line 398: I think you should specify, in every occasion, what you mean by cold water, by mentioning the temperatures. What is the temperature band utilized in veterinary studies, when you talk about cold water? The reader must be provided with a detailed literature review of the already tested cold temperatures in order to understand the importance of your results.
Lines 418-423: Is there another potential physiologic reason for the increased heart rate after isopropyl alcohol application, apart from the potential irritation or its aversive odor?
Lines 482-483: Which meta-analysis? Provide the reference.
Comments on the Quality of English Language
Minor editing required.
Author Response
Dear authors, I have carefully reviewed the current manuscript. The objective of the study was to assess the efficacy of 3 post exercise cooling methods. It seems an intresting study in a topic that even now we have no clear indications as for the first step proposed strategies for the decrease of the body temperature or for the proper cooling temperatures that should be used (at least in case of a heatstroke). Although someone may think that the current research questions could be easily examined utilizing not such a sophisticated or demanding study design, I have defined several issues that need to be addressed or explained and refer to the current study design.
Introduction section
- Line 92: fix the marks/units of Celsious degrees (not only here, but throughout the whole manuscript)
We have fixed it throughout the manuscript.
- Line 107-108: use a reference for such an emphatic statement
Several references have been added for this, thank you.
- Lines 112-115: "Predict"? This is not a proper way of presenting the hypothesis statement. Can you predict the outcome of the study? Here should be the hypothesis of the study, which should be a testable statement, and not a definite one.
See the edited lines, below.
This study examined the relative efficacy of three postexercise cooling methods (partial cool water immersion, isopropyl alcohol on paw pads, and passive cooling) in working dogs and working-dogs-in-training with exertional heat stress with the aim of finding the most effective and efficient way to reduce core body temperature post-exertion. Given the effectiveness of partial water immersion for cooling in humans as compared to passive cooling, it is likely that partial water immersion would also cool at a faster rate than passive cooling in dogs. However, it is yet unclear how the rate of cooling for water immersion would compare to the rate of cooling for another referenced but unexamined canine cooling method - use of isopropyl alcohol on paw pads – or how the rate of cooling for isopropyl alcohol application would compare to passive cooling. Temperature and heart rate were examined before and for 20 minutes after the cooling process to assess the impact of partial water immersion, isopropyl alcohol application to paw pads, and passive cooling on cooling rate and heart rate.
Material and Methods section
- The major problem of this section is the many missing (not mentioned) ARRIVE guidelines criteria.
The missing components of the ARRIVE guidelines have been added within the methods and materials sections.
- Which were the primary outcomes of the study? The average time that the body temperature returned to normal? I assume that the Heart Rate differences were the secondary outcomes of the study? They should be pre-specified and clearly mentioned.
The rate at which the temperature cooled during the time period, which was added to the paragraph at the end of the introduction. Heart rate was also mentioned as a secondary measure.
- What was the type of the study? Was it a pilot or a prospective study?
A subheading called study design and included the study type.
“2.2. Study Design
This study used a randomized crossover design. Blinding was not possible in this study due to the nature of the interventions.”
- There is no mention of the sample size calculation.
This has been added in the Participants section.
Sample size estimation was done in Glimmpse [29] based on a range of estimated standard deviations derived from a similar study [30]. A Hotelling-Lawley Trace with an alpha of 0.05 and a power of 0.8 with scaled standard deviations yields a sample size between 8 and 18. An effect size calculation done in G Power [31] based on an alpha of 0.05, a power set at 0.8 suggests that use of 11 participants in a crossover design can detect a medium effect size (estimated at d=0.5) with an approximate repeated measurements correlation above 0.65.
- What about inclusion/exclusion criteria? You do mention some inclusion criteria, but what about the exclusion criteria? Could you include a 15 years-old dog, for example?
Thank you, the exclusion criteria has been added. The sample for this study was a convenience sample obtained from the Penn Vet Working Dog Center population.
- You mention that this was a cross-over study (this is only mentioned in the Abstract). Which was the randomization plan? You do mention that the dogs were randomly assigned to each treatment (Lines 130-131), but which was the way of doing that?
A simple randomization was utilized with a random number generator. This has been added to the methods.
- I assume that there was no blinding throughout the procedure. However, that should have been mentioned, as well.
Correct, a section has been added to the text indicating that there was no blinding in the study.
Results section
- Which are the demographic characteristics of the dogs that participated in the study?
The demographic characteristics are listed in the supplemental material table.
- Lines 315-316: What is this temperature mentioned reffering to? This is mean and standard deviation of all dogs in the different study days and in the different treatments applied?
Thank you for your comment. A clarification statement has been added to these lines.
- Figure 1: What information does this figure provide? Timepoints 1-22 correspond to 1-22 minutes? When did the dog begin to exercise throughout this graph? At which timepoint? And when did the exercise stop? And at which timepoint was the 30second treatment applied? 22 points of data were provided. Were the dogs resting for that 22 min? Were they removed precisely at the 20 min point?
Thank you, the heart rate graph has been edited. The previous graph showed temperature averaged over approximately 2-min periods (leading to about 12 time points). The current graph shows temperature averaged over 1-min periods (20 time points). Temperature was measured for 2 more minutes than heart rate was measured due heart rate monitor equipment error - the heart rate collection began two minutes later than the temperature collection. Since the heart rate and temperature are not included in the same analysis, the full collected data for both heart rate and temperature were used in their respective analyses, even though temperature was collected for 2 additional minutes for temperature.
Discussion section
- Lines 386-390: You should mention that the study in horses does not utilize a water immersion. Mention the different strategies (sponging and pouring cold water) of either 9 or 31 degrees of tap water. Apart from the use of cooling (or not) water, there is no other resemblance to the current study design. It is true that it seems a safe first line defence, but as long as you review the literature, you should scrutinize the different approaches utilized.
This has been added.
- Lines 393-395: I agree that this is the result of Ref 19, but more recent studies (Flournoy et al. 2003) propose that low temperatures (ice water baths) are contraindicated.
Even more recent studies in humans suggest that low temperature baths are acceptable (Nye, 2016); the data is not clear at this time and requires further investigation. The setting of which these cooling methods are attempted in addition to the stage of hyperthermia (stress, exhaustion, stroke) is an important factor as well.
- Line 398: I think you should specify, in every occasion, what you mean by cold water, by mentioning the temperatures. What is the temperature band utilized in veterinary studies, when you talk about cold water? The reader must be provided with a detailed literature review of the already tested cold temperatures in order to understand the importance of your results.
Temperatures were added for all studies in which they were provided.
- Lines 418-423: Is there another potential physiologic reason for the increased heart rate after isopropyl alcohol application, apart from the potential irritation or its aversive odor?
This is an interesting posed question, but very unlikely. We do not believe there is a physiologic reason for the isopropyl alcohol to alter heart rate. There is no literature to support absorption through the paw pads or integument, and none of the participants consumed the liquid.
- Lines 482-483: Which meta-analysis? Provide the reference.
This is Nye, 2016, which has been added.
Round 2
Reviewer 3 Report
Comments and Suggestions for Authors
Thank you for your feedback.
Good luck.
Comments on the Quality of English LanguageMinor editing required